# Evaluation of Schools of University and Analysis of Obstacle Factors under the Background of High-Quality Development

**Huiping Chen**

School of Economics and Management, Nantong University, Nantong 226019, China; chpgzb@ntu.edu.cn

**Abstract:** Currently, promoting the high-quality development (HQD) of higher education is an urgent need of the country, which is the requirement for achieving stable and sustainable development. As the main bodies of the university, the development levels of the schools directly affect universities' development. To improve universities' comprehensive strength, this paper proposes an integrated framework for HQD evaluation and obstacle-factor analysis for schools. To address this problem, combining the management by objectives (MBO) constructs an HQD evaluation model of schools of university covering different aspects. Meanwhile, the obstacle diagnosis model is utilized to analyze the main obstacle factors that restrict the improvement of HQD level of the schools. Moreover, taking into account the vagueness and imprecision in real life, index weights and expert weights are determined through the intuitionistic fuzzy set (IFS). Finally, a practical problem was chosen to illustrate the efficiency and applicability of the proposed framework and some suggestions from different perspectives are given according to the analysis of results.

**Keywords:** high-quality development; obstacle diagnosis; evaluation; intuitionistic fuzzy set; weight

## 1. Introduction

In response to the challenges and opportunities presented by the world situation, the just-held 20th National Congress of the Communist Party of China called for high-quality development (HQD) transformation in economic, social, environmental, resource and other aspects. Generally, education is related to the growth of individuals and the rise and fall of the nation and it is the foundation of major projects for people's livelihood and the prosperity of the country [1]. In a nutshell, education is the foundation of people's livelihood and the foundation of a country. Therefore, education must first achieve HQD. In China, higher education has developed rapidly in recent years, and it plays a significant part in the development of the whole society and economy. In order to ensure high quality of development, reform of higher education is the only way to adapt to the trend of social development in the new era. Higher education evaluation can provide a clear direction for comprehensive reform, so as to facilitate the HQD of colleges and universities.

So far, higher education evaluation has received substantial interest and more attention. Astin and Antonio [2] put forward the evaluation index system of higher education development level. The cost-effectiveness evaluation method of higher education institutions was considered based on the compound data envelopment analysis model [3]. By combining data envelopment analysis and analytical hierarchy process, application research on the efficiency assessment of universities was developed [4]. The slacks-based measure model was used to evaluate the performance of universities [5]. Based on the grey analytic hierarchy process, Sun evaluated the situation of faculty and staff and analyzed the initiative and enthusiasm of faculty and staff in the construction of "double first-class universities" [6]. Huang and his partners used the fuzzy-ANP comprehensive evaluation model to create an index system for building high-level universities with Jiangsu Province features [7]. As the entities of the university, the improvement of the comprehensive strength of the schools will determine the HQD level of the university. Therefore, we

should take the schools of the university as the objects of educational evaluation and take maximum advantage of the evaluation baton, so as to continuously activate the vitality and power of the universities. Combining the theory of the management by objectives (MBO) with the management of secondary colleges, Liu et al. introduced the method of quantitative assessment and overall calculation of secondary colleges [8]. On the basis of the theory and method of MBO, the performance evaluation system of secondary colleges in universities was constructed [9]. Hu [10] conducted an empirical analysis on the development level of higher education in indifferent regions of China via factor analysis to explore and explain the reasons for the regional differences in the development of higher education. Xun with his collaborators designed a three-dimensional evaluation system of colleges and schools led by goal achievement [11].

Admittedly, it is not easy to describe or estimate the possible significance of HQD in view of the highly intricate and imprecise ideas. In the process of evaluation, due to the disturbance of internal and external complex factors and the limitation of the human cognitive level, the obtained information is often uncertain and fuzzy. Furthermore, it is difficult for decision makers to give accurate evaluation values under the complex evaluation environment. In order to better describe the uncertainty of evaluation values, Zadeh proposed the concept of Fuzzy Sets (FSs) [12]. It has been extensively applied to the comprehensive evaluation [13–15].

As an improvement of FSs, the intuitionistic fuzzy set (IFS) [16] can better describe the complexity and fuzziness of information, depicted by the membership information, non-membership information and hesitation information. Same as FSs, IFSs have been extensively used by many researchers in different fields, such as decision making, pattern recognition, etc. IFSs were used to evaluate students and a classification algorithm was proposed [17]. Bureva and his coworkers proposed a novel approach based on IFSs for evaluation of universities and determination of possible dependencies between the evaluation criteria [18]. Based on the complex proportional assessment method and IFSs, Mishra and his coworkers [19] decided the ranking of bio-energy production technology alternatives. Dong [20] combined the multiple attribute decision making method with interval-valued IFSs to evaluate the mental health status of poverty-stricken college students at the present age. A novel integrated multi-criteria decision-making model was put forward to evaluate smart city development based on the intuitionistic fuzzy analytical hierarchy process and the intuitionistic fuzzy decision-making trial and evaluation laboratory [21]. Yuan and Zheng [22] applied the new intuitionistic fuzzy entropy to evaluate the regional collaborative innovation capability, which comprehensively considered the deviation between membership and non-membership and the influence of hesitation. Wu et al. proposed a new method for linguistic intuitionistic fuzzy group decision-making to evaluate land reclamation schemes in mining areas [23]. Based on the interval intuitionistic fuzzy theory, Qi designed and developed the teaching evaluation system for the evaluation of classroom teaching quality in universities [24].

In particular, the obstacle diagnosis model evolved on the basis of the comprehensive evaluation model. It is a mathematical model that can fully explore the obstacles and find out the main obstacle factors that limit the development of things. Hu et al. [25] constructed the quality evaluation index system of economic development and utilized the obstacle diagnosis model for empirical analysis. Through the obstacle diagnosis model, the main obstacle factors restricting the improvement of urban ecological level were obtained and studied [26]. Via the obstacle diagnosis model, Jiang et al. [27] obtained and analyzed the main obstacle factors restricting the improvement of the HQD level. The existing higher education evaluation system has established a positive incentive mechanism, which has played a guiding role in the development of schools of the universities. Guo and Chen analyzed the HQD level and main restrictive factors of rural e-commerce in China under the new development concept by using an obstacle factor diagnosis model [28]. An obstacle degree model was used to investigate the key factors functioning as obstacles to food security [29].

However, evaluation activities have sometimes only ranked and given quantitative indicators, directly given the calculation of the total score of the diagnostic report and failed to put forward targeted practical strategies for HQD of schools of the universities. This paper attempts to construct an integrated model to objectively measure and comprehensively evaluate the HQD level of the schools in universities. Moreover, we employ the obstacle diagnosis model to help identify the obstacles affecting their HQD. One typical example is selected to analyze the results of HQD evaluation and obstacle diagnosis. Some suggestions for guidance for the development of HQD strategies are proposed.

The rest of this paper is structured as follows. Section 2 introduces some fundamental concepts related to IFSs. Subsequently, an integrated framework is proposed to calculate the HQD level of schools of the university and establish the obstacle diagnosis model in Section 3. Meanwhile, IFSs are employed to evaluate the expert weights and the index weights. To reveal the practicability and feasibility, Section 4 implements the proposed framework in a case study of HQD evaluation of schools in a university. After analyzing the evaluation and obstacle diagnosis results, some suggestions are presented to push the development of the university to a new stage. Section 5 concludes the work.

## 2. Preliminaries

Here, some basic concepts about IFSs are reviewed.

**Definition 1** ([16]). *An IFS $A$ on a fixed set $X$ is defined with the form $A = \{< x, \mu_A(x), \nu_A(x) > | x \in X\}$, where its membership degree $\mu_A$ and non-membership degree $\nu_A$ satisfy the following conditions*

$$\mu_A : X \to [0,1], \quad \nu_A : X \to [0,1], \quad 0 \le \mu_A(x) + \nu_A(x) \le 1. \tag{1}$$

Let $\pi_A(x) = 1 - \mu_A(x) - \nu_A(x)$; it describes the degree of hesitation of $x$ to $A$. Obviously, $\pi_A(x) \in [0,1]$. For convenience, call $\xi = (\mu_\xi, \nu_\xi)$ the intuitioistic fuzzy number (IFN), which satisfies $\mu_\xi + \nu_\xi \le 1$ and $\mu_\xi \in [0,1]$, $\nu_\xi \in [0,1]$ [30]. $s(\xi) = \mu_\xi - \nu_\xi$ and $h(\xi) = \mu_\xi + \nu_\xi$ refer to score function and accuracy function, respectively.

**Definition 2** ([31]). *Let $\xi_i = (\mu_i, \nu_i)$ $(i = 1, 2, \cdots, n)$ be $n$ IFNs. The intuitionistic fuzzy weighted average (IFWA) and the intuitionistic fuzzy weighted geometric (IFWG) operators are given by*

$$\text{IFWA}(\xi_1, \xi_2, \cdots, \xi_n) = \bigoplus_{i=1}^{n} \omega_i \xi_i = \left(1 - \prod_{i=1}^{n}(1-\mu_i)^{\omega_i}, \prod_{i=1}^{n}(\nu_i)^{\omega_i}\right), \tag{2}$$

$$\text{IFWG}(\xi_1, \xi_2, \cdots, \xi_n) = \bigotimes_{i=1}^{n} \omega_i \xi_i = \left(\prod_{i=1}^{n}(\mu_i)^{\omega_i}, 1 - \prod_{i=1}^{n}(1-\nu_i)^{\omega_i}\right), \tag{3}$$

*where $\omega_i$ $(i = 1, 2, \cdots, n)$ is the weight of $\xi_i$ such that $\omega_i \in [0,1]$ and $\sum\limits_{i=1}^{n} \omega_i = 1$.*

## 3. The Proposed Integrated Framework

In the current section, an integrated framework with the comprehensive evaluation and the obstacle analysis is established under the intuitionistic fuzzy environment. The working procedures of the integrated framework are discussed as follows.

### 3.1. Calculation of HQD Level of the Schools of University

Here, on the one hand, the theory related to IFSs is applied to obtain the weights of experts and indicators. On the other hand, MBO is used to calculate the HQD level of the schools of a university. Details are as follows.

**Step 1:** Construct an evaluation index system. That is, define the evaluation objects and select the appropriate evaluation indexes.

There are $m$ evaluation objects (the schools of university) $C = \{C_1, C_2, \cdots, C_m\}$. Consider a committee of $l$ experts $E = (e_1, e_2, \cdots, e_l)$, which has been formed to determine evaluation indexes. Evaluation indexes and their weight information are described in Table 1.

**Table 1.** Evaluation index system of the schools of a university.

| Level Indicators | Weight of Level Indicators | Secondary Indicators | Weight of Secondary Indicators |
|:---:|:---:|:---:|:---:|
| $A_1$ | $\omega_1$ | $B_1^1$ <br> ... <br> $B_{l_1}^1$ | $w_1^1$ <br> ... <br> $w_{l_1}^1$ |
| $A_2$ | $\omega_2$ | $B_1^2$ <br> ... <br> $B_{l_2}^2$ | $w_1^2$ <br> ... <br> $w_{l_2}^2$ |
| . <br> . <br> . | . <br> . <br> . | . <br> . <br> . | . <br> . <br> . |
| $A_n$ | $\omega_n$ | $B_1^n$ <br> ... <br> $B_{l_n}^n$ | $w_1^n$ <br> ... <br> $w_{l_n}^n$ |

**Step 2:** Assign the weight of each expert.

The importance of the $e_k$ is estimated through a linguistic term expressed per IFN. Table 2 shows the linguistic scale and its corresponding IFN used for evaluating the weight of the $e_k$.

**Table 2.** Linguistic scale for assessing the importance of $e_k$ and index.

| Linguistic Terms | IFNs |
|:---:|:---:|
| Beginner/Very Unimportant | (0.1,0.9,0) |
| Practitioner/Unimportant | (0.35,0.60,0.05) |
| Proficient/Medium | (0.50,0.45,0.05) |
| Expert/Important | (0.75,0.20,0.05) |
| Master/Very Important | (0.9,0.1,0) |

Let $E(\mu_{e_k}, \nu_{e_k}, \pi_{e_k})$ be an IFN for rating $e_k$. Then, the weight $\lambda_k$ regarding $e_k$ is calculated as

$$\lambda_k = \frac{\left(\mu_{e_k} + \pi_{e_k}\left(\frac{\mu_{e_k}}{\mu_{e_k} + \nu_{e_k}}\right)\right)}{\sum\limits_{t=1}^{l}\left(\mu_{e_t} + \pi_{e_t}\left(\frac{\mu_{e_t}}{\mu_{e_t} + \nu_{e_t}}\right)\right)}, \tag{4}$$

where $\lambda_k \geq 0$, $\sum\limits_{k=1}^{l} \lambda_k = 1$.

**Step 3:** Delineate the preferences of the indexes.

As is well known, experts might give variant preferences about the same indexes. They utilize the linguistic class shown in Table 2 to judge the preferences of indexes. Subsequently, all preferences need to be analyzed and conglomerated into one, so as to deduce the weight information of the indexes.

Let $\xi_i^k = (\mu_i^k, \nu_i^k, \pi_i^k)$ and $\xi_j^{ik} = (\mu_j^{ik}, \nu_j^{ik}, \pi_j^{ik})$ be IFNs assigned to first-level index $A_i$ and second-level index $B_j^i$ by the expert $e_k$, respectively. Via Equation (2), the opinions of experts about $\xi_i^k$ and $\xi_j^{ik}$ are integrated as follows.

$$\xi_i = IFWA(\xi_i^1, \xi_i^2, \cdots, \xi_i^l) = \bigoplus_{k=1}^{l} \lambda_k \xi_i^k = \left(1 - \prod_{k=1}^{l}(1 - \mu_i^k)^{\lambda_k}, \prod_{k=1}^{l}(\nu_i^k)^{\lambda_k}\right), \tag{5}$$

$$\xi_j^i = IFWA(\xi_j^{i1}, \xi_j^{i2}, \cdots, \xi_j^{il}) = \bigoplus_{k=1}^{l} \lambda_k \xi_j^{ik} = \left(1 - \prod_{k=1}^{l}(1 - \mu_j^{ik})^{\lambda_k}, \prod_{k=1}^{l}(\nu_j^{ik})^{\lambda_k}\right), \tag{6}$$

where $\xi_i = (\mu_i, \nu_i, \pi_i)$, $\xi_j^i = (\mu_j^i, \nu_j^i, \pi_j^i)$ are aggregated IFNs of $\xi_i^k$ and $\xi_j^{ik}$, respectively.

Next, the corresponding weight of the first-level index $A_i$ is obtained as:

$$\omega_i = \frac{\left(\mu_i + \pi_i(\frac{\mu_i}{\mu_i + \nu_i})\right)}{\sum\limits_{t=1}^{n}\left(\mu_t + \pi_t(\frac{\mu_t}{\mu_t + \nu_t})\right)}, \tag{7}$$

where $\omega_i \geq 0$, $\sum\limits_{i=1}^{n} \omega_i = 1$.

In the same way, the weight of the second-level index $B_j^i$ is obtained by

$$w_j^i = \frac{\left(\mu_j^i + \pi_j^i(\frac{\mu_j^i}{\mu_j^i + \nu_j^i})\right)}{\sum\limits_{t=1}^{l_i}\left(\mu_t^i + \pi_t^i(\frac{\mu_t^i}{\mu_t^i + \nu_t^i})\right)}, \tag{8}$$

where $w_j^i \geq 0$, $\sum\limits_{j=1}^{l_i} w_j^i = 1$.

**Step 4:** Determine normalized evaluation matrix.

**Step 4.1:** Obtain the evaluation values.

Let $y_{tj}^i$ and $Y_{tj}^i$ be completion value and target value of the school $C_t$ under the indicator index $B_j^i$, respectively. Then, the score $x_{tj}^i$ of $C_t$ under index $B_j^i$ is

$$x_{tj}^i = \frac{y_{tj}^i}{Y_{tj}^i}. \tag{9}$$

**Step 4.2:** Standardize the scores:

$$a_{tj}^i = \begin{cases} \frac{x_{tj}^i \times 100}{\max_{t=1}^{m}(x_{tj}^i)}, & \max_{t=1}^{m}(x_{tj}^i) > 1, \\ x_{tj}^i \times 100, & \max_{t=1}^{m}(x_{tj}^i) \leq 1. \end{cases} \tag{10}$$

Therefore, the normalized matrix $AX^i = (a_{tj}^i)_{m \times l_i}$ is obtained.

**Step 5:** Synthesize the scores of each level indicator. Let $H_i^t$ be the score of $C_t$ under $A_i$. By the weighted average and the weight $w_j^i$ of $B_j^i$ obtained in Step 3, the value of each level indicator is calculated.

$$H_i^t = \sum_{j=1}^{l_i} w_j^i \times a_{tj}^i, \tag{11}$$

where $l_i$ is the number of evaluation indicators of the $i$th level indicator $A_i$.

**Step 6:** Integrate and obtain the HQD level. According to the weight $\omega^i$ of $A_i$ in Step 3, the HQD level of the school $C_t$ is the weighted average of the $n$ values calculated above.

$$H^t = \sum_{i=1}^{n} \omega^i \times H_i^t. \tag{12}$$

**Step 7:** Rank the schools according to the $H^t$. The larger the $H^t$, the better the school $C_t$.

### 3.2. Establish the Obstacle Diagnosis Model

The evaluation of the schools of university aims to judge the HQD level. To better formulate countermeasures and suggestions for pushing forward the development of schools, this section makes use of the obstacle diagnosis model to achieve two goals. One is to obtain all indexes' degree of obstacles that restrict schools' development. The other is to identify obstacles' factors and provide a targeted decision-making basis for the development of the university's schools. The obstacle degree $S_{tj}^i$ of the $C_t$ under the indicator $B_i^i$ is calculated as follows:

$$S_{tj}^i = \frac{(1 - a_{tj}^i) \times (w_j^i \times \omega_i)}{\sum\limits_{k=1}^{l_i} (1 - a_{tk}^i) \times (w_k^i \times \omega_i)}, \quad S_{tj}^i \in [0, 1]. \tag{13}$$

Obviously, the greater the $S_{tj}^i$, the more obstructive the indicator $B_j^i$.

## 4. Case Study: Evaluation and Analysis

### 4.1. Study Object and Data Source

In June 2020, the Ministry of Education of China officially released a list of colleges and universities. It indicated that there were 2740 regular colleges and universities in China. The distribution of colleges and universities in each province could be seen from Figure 1. Jiangsu Province alone had 167 universities including 78 undergraduate colleges and 89 junior colleges. Let me put it another way, Jiangsu Province had the largest number of undergraduate colleges and universities in China. As a large province of education, the quality of such a large educational group in Jiangsu Province is directly related to the quality of its higher education system. As the entities of a university, the improvement of the comprehensive strength of the schools determines the HQD level of the university. Consequently, schools should be regarded as the objects of educational evaluation to give full play to the role of evaluation baton, so as to continuously activate the vitality and strength of the schools. Nantong University is a representative comprehensive university in the developing stage in Jiangsu Province. A study of HQD evaluation of Nantong University will make the proposed method understandable and specific.

In order to make the evaluation results more reasonable and scientific, the experts are assigned different weights according to their different levels and experience. Here, four experts are integrated in the group and their relative importance is shown in Table 3. In order to promote the HQD of colleges and universities, Jiangsu Province has formulated the measures for the implementation of a comprehensive assessment of local colleges and universities and has also formulated assessment indicators. Therefore, considering the comprehensive evaluation index of local colleges and universities in Jiangsu Province and the development goal of Nantong University, the experts set up the HQD evaluation index system of Nantong University as completely as possible. It contains four first-level indicators and 15 second-level indicators, as shown in Table 4.

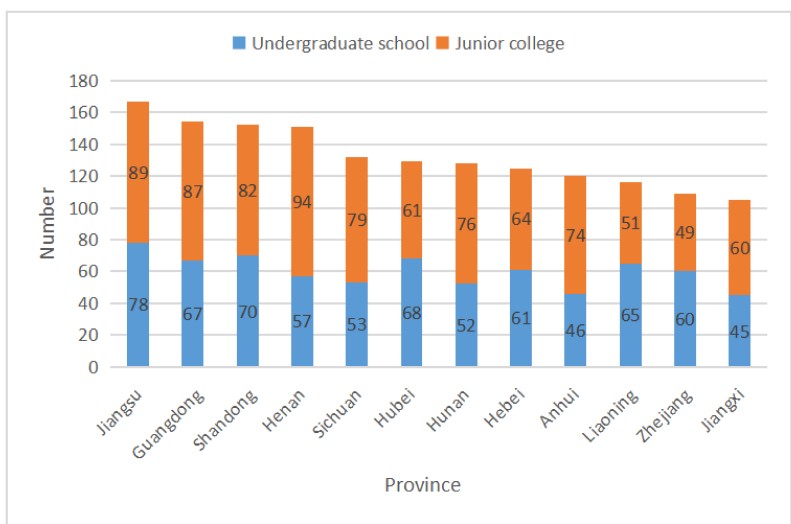

**Figure 1.** Geographical distribution map of the number of regular universities exceeding 100 in 2020 (Unit: Institute). Notes: (1) Data as of: 30 June 2020; (2) Source: Ministry of Education of China.

**Table 3.** The relative importance of experts.

| | **Linguistic Terms** | **IFNs** |
|---|---|---|
| $e_1$ | Master | (0.9,0.1,0) |
| $e_2$ | Proficient | (0.50,0.45,0.05) |
| $e_3$ | Expert | (0.75.,0.20,0.05) |
| $e_4$ | Master | (0.9,0.1,0) |

**Table 4.** The HQD evaluation system.

| First-Level Index Layer | Second-Level Index Layer |
|---|---|
| $A_1$ Talent and teaching staff construction | $B_1^1$: Percentage of full-time teachers with a doctoral degree<br>$B_2^1$: Proportion of full-time teachers with more than six months of overseas exchange<br>$B_3^1$: Percentage of full-time teachers with more than one year of overseas study, research and teaching experience<br>$B_4^1$: Number of teachers who have studied in world-class universities for more than one year<br>$B_5^1$: The number of newly added national key talents<br>$B_6^1$: The number of newly introduced overseas doctoral degree teachers<br>$B_7^1$: The number of newly added provincial key talents |
| $A_2$ Undergraduate education and quality engineering construction | $B_1^2$: The percentage of professors teaching undergraduate courses<br>$B_2^2$: National competition awards<br>$B_3^2$: The number of teaching and research activities organized by the State Education Commission<br>$B_4^2$: The newly added provincial colleges and universities outstanding graduation design (thesis) selection award<br>$B_5^2$: Undergraduate research situation<br>$B_6^2$: The number of college students registered for innovation and entrepreneurship competition<br>$B_7^2$: Employment of undergraduate graduates<br>$B_8^2$: Student management and services |
| $A_3$ Postgraduate training and discipline construction | $B_1^3$: Discipline construction<br>$B_2^3$: The number of new graduate students with overseas study experience<br>$B_3^3$: Year-end total employment rate of graduate students<br>$B_4^3$: Total initial employment rate of graduate students<br>$B_5^3$: Graduate Admissions—voluntary admission rate<br>$B_6^3$: The proportion of graduate students enrolled from dual-top universities<br>$B_7^3$: Proportion of graduates from our university who are recommended to study for master's degree without examination<br>$B_8^3$: Postgraduate dissertation sampling pass rate |

**Table 4.** *Cont.*

| First-Level Index Layer | Second-Level Index Layer |
|---|---|
| $A_4$ Scientific research and social service | $B_1^4$: Newly added national natural science research project<br>$B_2^4$: New vertical and horizontal research funds<br>$B_3^4$: Number of newly published high quality papers<br>$B_4^4$: Increased number of high-level academic works published by authoritative and famous publishing houses<br>$B_5^4$: New provincial and ministerial science and technology achievement awards<br>$B_6^4$: Status of invention patents<br>$B_7^4$: Production, education and research platform construction<br>$B_8^4$: Social donations |

Additionally, each expert gave the grade of preferences of the first layer and second layer depicted as a linguistic class, as shown in Tables 5 and 6, respectively.

**Table 5.** The importance of first layer.

| | $e_1$ | $e_2$ | $e_3$ | $e_4$ | Weights |
|---|---|---|---|---|---|
| $A_1$ | (0.75,0.20,0.05) | (0.50,0.45,0.05) | (0.50,0.45,0.05) | (0.75,0.20,0.05) | 0.2378 |
| $A_2$ | (0.9,0.1,0) | (0.75,0.20,0.05) | (0.75,0.20,0.05) | (0.9,0.1,0) | 0.2926 |
| $A_3$ | (0.50,0.45,0.05) | (0.35,0.50,0.05) | (0.50,0.45,0.05) | (0.50,0.45,0.05) | 0.1728 |
| $A_4$ | (0.9,0.1,0) | (0.9,0.1,0) | (0.9,0.1,0) | (0.75,0.20,0.05) | 0.2968 |

**Table 6.** The importance of second layer.

| | $e_1$ | $e_2$ | $e_3$ | $e_4$ | Weights |
|---|---|---|---|---|---|
| $B_1^1$ | (0.9,0.1,0) | (0.9,0.1,0) | (0.9,0.1,0) | (0.9,0.1,0) | 0.2530 |
| $B_2^1$ | (0.1,0.9,0) | (0.35,0.50,0.05) | (0.1,0.9,0) | (0.35,0.50,0.05) | 0.0692 |
| $B_3^1$ | (0.35,0.50,0.05) | (0.35,0.50,0.05) | (0.35,0.50,0.05) | (0.35,0.50,0.05) | 0.1158 |
| $B_4^1$ | (0.1,0.9,0) | (0.50,0.45,0.05) | (0.1,0.9,0) | (0.1,0.9,0) | 0.0528 |
| $B_5^1$ | (0.50,0.45,0.05) | (0.50,0.45,0.05) | (0.50,0.45,0.05) | (0.50,0.45,0.05) | 0.1480 |
| $B_6^1$ | (0.50,0.45,0.05) | (0.50,0.45,0.05) | (0.50,0.45,0.05) | (0.50,0.45,0.05) | 0.1480 |
| $B_7^1$ | (0.75,0.20,0.05) | (0.50,0.45,0.05) | (0.75,0.20,0.05) | (0.75,0.20,0.05) | 0.2132 |
| $B_1^2$ | (0.50,0.45,0.05) | (0.1,0.9,0) | (0.1,0.9,0) | (0.1,0.9,0) | 0.0656 |
| $B_2^2$ | (0.9,0.1,0) | (0.9,0.1,0) | (0.9,0.1,0) | (0.9,0.1,0) | 0.2398 |
| $B_3^2$ | (0.1,0.9,0) | (0.50,0.45,0.05) | (0.1,0.9,0) | (0.1,0.9,0) | 0.0500 |
| $B_4^2$ | (0.35,0.50,0.05) | (0.50,0.45,0.05) | (0.35,0.50,0.05) | (0.35,0.50,0.05) | 0.1159 |
| $B_5^2$ | (0.35,0.50,0.05) | (0.35,0.50,0.05) | (0.35,0.50,0.05) | (0.50,0.45,0.05) | 0.1200 |
| $B_6^2$ | (0.35,0.50,0.05) | (0.35,0.5,0.05) | (0.35,0.50,0.05) | (0.35,0.50,0.05) | 0.1097 |
| $B_7^2$ | (0.50,0.45,0.05) | (0.75,0.20,0.05) | (0.35,0.50,0.05) | (0.50,0.45,0.05) | 0.1506 |
| $B_8^2$ | (0.50,0.45,0.05) | (0.50,0.45,0.05) | (0.50,0.45,0.05) | (0.35,0.50,0.05) | 0.1484 |
| $B_1^3$ | (0.9,0.1,0) | (0.9,0.1,0) | (0.9,0.1,0) | (0.9,0.1,0) | 0.2056 |
| $B_2^3$ | (0.50,0.45,0.05) | (0.50,0.45,0.05) | (0.50,0.45,0.05) | (0.50,0.45,0.05) | 0.1202 |
| $B_3^3$ | (0.50,0.45,0.05) | (0.50,0.45,0.05) | (0.50,0.45,0.05) | (0.50,0.45,0.05) | 0.1202 |
| $B_4^3$ | (0.35,0.50,0.05) | (0.35,0.50,0.05) | (0.35,0.50,0.05) | (0.35,0.50,0.05) | 0.0941 |
| $B_5^3$ | (0.50,0.45,0.05) | (0.1,0.9,0) | (0.35,0.50,0.05) | (0.35,0.50,0.05) | 0.0924 |
| $B_6^3$ | (0.35,0.50,0.05) | (0.35,0.50,0.05) | (0.35,0.50,0.05) | (0.35,0.50,0.05) | 0.0941 |
| $B_7^3$ | (0.75,0.20,0.05) | (0.50,0.45,0.05) | (0.50,0.45,0.05) | (0.50,0.45,0.05) | 0.1426 |
| $B_8^3$ | (0.35,0.50,0.05) | (0.50,0.45,0.05) | (0.75,0.20,0.05) | (0.1,0.9,0) | 0.1308 |
| $B_1^4$ | (0.75,0.20,0.05) | (0.75,0.20,0.05) | (0.75,0.20,0.05) | (0.75,0.20,0.05) | 0.1696 |
| $B_2^4$ | (0.9,0.1,0) | (0.9,0.1,0) | (0.9,0.1,0) | (0.9,0.1,0) | 0.1933 |
| $B_3^4$ | (0.50,0.45,0.05) | (0.50,0.45,0.05) | (0.50,0.45,0.5) | (0.50,0.45,0.05) | 0.1130 |
| $B_4^4$ | (0.1,0.9,0) | (0.1,0.9,0) | (0.1,0.9,0) | (0.1,0.9,0) | 0.0215 |
| $B_5^4$ | (0.50,0.45,0.05) | (0.50,0.45,0.05) | (0.50,0.45,0.5) | (0.50,0.45,0.05) | 0.1130 |
| $B_6^4$ | (0.75,0.20,0.05) | (0.9,0.1,0) | (0.9,0.1,0) | (0.9,0.1,0) | 0.1884 |
| $B_7^4$ | (0.1,0.9,0) | (0.1,0.9,0) | (0.1,0.9,0) | (0.1,0.9,0) | 0.0215 |
| $B_8^4$ | (0.75,0.20,0.05) | (0.75,0.20,0.05) | (0.75,0.20,0.05) | (0.75,0.20,0.05) | 0.1797 |

Considering the rationality of evaluation, it is necessary to formulate differentiated evaluation criteria for the characteristics of different disciplines such as arts and sciences. Here, we take 2020 as the time point for research and select 11 schools of science and engineering in Nantong University to study the comprehensive evaluation, so as to further clarify HQD around the schools' differences and imbalance. The basic index data comes from the comprehensive examination results of local ordinary universities in Jiangsu Province and the public and statistical information of Nantong University.

### 4.2. Evaluation of HQD Level

In this section, we use the proposed method in Section 3.1 to obtain the HQD level of 11 schools and sort them accordingly. Calculation procedures for the HQD level of 11 schools in Nantong University are described as follows:

**Step 1:** Obtain the weights of experts. According to Equation (4) and preference information in Table 3, the weights of the experts can be obtained, namely

$$\lambda = (\lambda_1, \lambda_2, \lambda_3, \lambda_4) = (0.2889, 0.1689, 0.2533, 0.2889).$$

**Step 2:** Calculate the weights of first-level and second-level indicators, respectively.

Make use of Equations (5)–(8) in turn in Section 3.2, the weights of first-level and second-level indicators are obtained, see details in Tables 5 and 6, respectively.

**Step 3:** Obtain normalized evaluation matrix and synthesized vales under the first level.

According to the target value and completion value of each secondary indicator for each school in 2020, we obtain the score of each school under each indicator. Subsequently, utilizing the method given in Section 3.1, we obtain the HQD level of each first-level indicator and their rankings are shown in Table 7.

**Step 4:** Rank the schools.

Using Equation (12), HQD level and ranking of the 11 schools are determined. See Table 7 for details.

**Table 7.** HQD level and ranking of schools.

| Schools | $A_1$ Scores | $A_1$ Rank | $A_2$ Scores | $A_2$ Rank | $A_3$ Scores | $A_3$ Rank | $A_4$ Scores | $A_4$ Rank | Comprehensive Scores | Comprehensive Rank |
|---|---|---|---|---|---|---|---|---|---|---|
| $C_1$ | 75.99 | 5 | 69.02 | 5 | 71.06 | 4 | 55.19 | 9 | 66.93 | 7 |
| $C_2$ | 97.36 | 1 | 68.08 | 6 | 65.78 | 9 | 73.92 | 2 | 76.38 | 1 |
| $C_3$ | 68.72 | 10 | 61.79 | 9 | 72.08 | 3 | 60.24 | 6 | 64.76 | 9 |
| $C_4$ | 79.92 | 4 | 73.90 | 4 | 70.96 | 5 | 71.31 | 3 | 74.05 | 3 |
| $C_5$ | 66.51 | 11 | 78.97 | 2 | 72.31 | 2 | 59.53 | 7 | 69.09 | 5 |
| $C_6$ | 69.61 | 8 | 83.72 | 1 | 69.74 | 6 | 67.52 | 4 | 73.14 | 4 |
| $C_7$ | 85.20 | 2 | 67.62 | 7 | 73.54 | 1 | 78.34 | 1 | 76.01 | 2 |
| $C_8$ | 80.98 | 3 | 75.80 | 3 | 63.76 | 10 | 52.73 | 10 | 68.10 | 6 |
| $C_9$ | 74.99 | 6 | 60.34 | 11 | 69.19 | 7 | 51.10 | 11 | 62.61 | 10 |
| $C_{10}$ | 68.73 | 9 | 62.32 | 8 | 68.13 | 8 | 65.14 | 5 | 65.68 | 8 |
| $C_{11}$ | 71.90 | 7 | 60.58 | 10 | 59.54 | 11 | 58.30 | 8 | 62.41 | 11 |
| Average value | 76.36 | / | 69.29 | / | 68.74 | / | 63.03 | / | 69.01 | / |

From Table 7, the order of the HQD level of schools is $C_2 \succ C_7 \succ C_4 \succ C_6 \succ C_5 \succ C_8 \succ C_1 \succ C_{10} \succ C_3 \succ C_9 \succ C_{11}$. Obviously, $C_2$ is the best one. However, having said that, there are some problems in this case. The ranking of $C_2$ under the index $A_3$ is relatively low. It points out the direction of future efforts for the school $C_2$. Table 7 also shows that most schools only have a weak HQD level under a certain first-level indicator. Although the HQD level of schools $C_9, C_{10}, C_{11}$ is relatively balanced in each first-level index, their level

is not very high. From the average of the HQD level of each first-level indicator, the order is $A_1 \succ A_2 \succ A_3 \succ A_4$. It implies that the school as a whole is the weakest in scientific research and social service, followed by graduate training and discipline construction.

To further understand the HQD level of each school, we select the base data from 2019 to 2021 and use the proposed method in Section 3.1 to calculate the changes under the first-level indicators and HOD level of the schools. This information is illustrated in Figure 2 and Table 8.

Considering the combination of four layers, the HQD level expresses the overall HQD level of first level. As shown in Figure 2, it is easy to find that the HQD level of all schools exhibits a fluctuating upward trend from 2019 to 2021. As can be seen from Figure 2, the development level of each indicator is on the whole on the rise from 2019 to 2020. However, most of them went down from 2020 to 2021. This is mainly due to the impact of the pandemic.

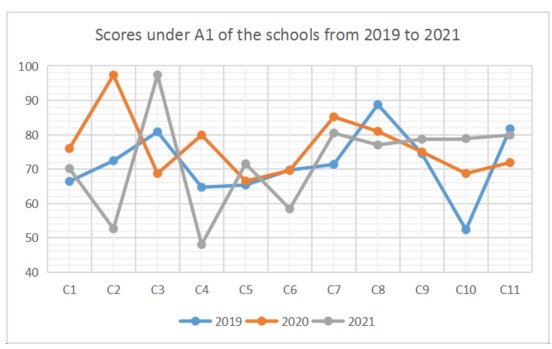
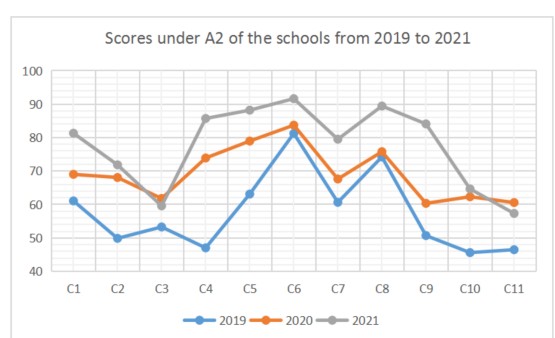

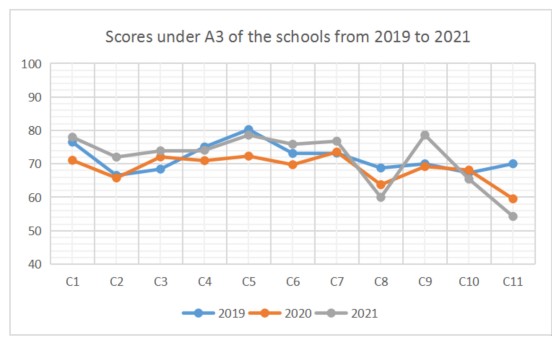
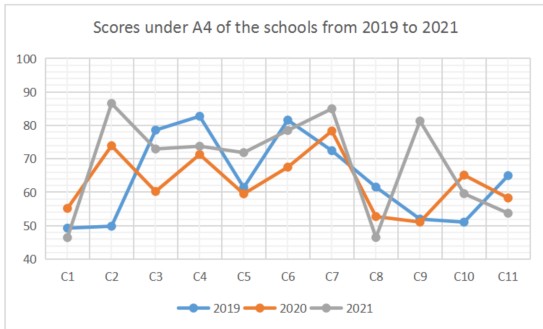

**Figure 2.** Scores of schools for different indexes from 2019 to 2021.

**Table 8.** HQD level and ranking of schools.

| Schools | 2019 | | 2020 | | 2021 | |
|---|---|---|---|---|---|---|
| | Scores | Rank | Scores | Rank | Scores | Rank |
| $C_1$ | 61.51 | 8 | 66.93 | 7 | 61.48 | 9 |
| $C_2$ | 58.11 | 10 | 76.38 | 1 | 66.99 | 5 |
| $C_3$ | 69.97 | 3 | 64.76 | 9 | 66.33 | 7 |
| $C_4$ | 66.68 | 5 | 74.05 | 3 | 66.92 | 6 |
| $C_5$ | 66.11 | 6 | 69.09 | 5 | 71.36 | 4 |
| $C_6$ | 77.21 | 1 | 73.14 | 4 | 71.90 | 3 |
| $C_7$ | 68.87 | 4 | 76.01 | 2 | 73.72 | 2 |
| $C_8$ | 72.98 | 2 | 68.10 | 6 | 61.78 | 8 |
| $C_9$ | 60.07 | 9 | 62.61 | 10 | 74.02 | 1 |
| $C_{10}$ | 52.58 | 11 | 65.68 | 8 | 59.63 | 10 |
| $C_{11}$ | 64.42 | 7 | 62.41 | 11 | 53.97 | 11 |

To reflect the HQD level of schools intuitively, we create seven levels: no levels $(0)$, very poor $((0, 20))$, poor $([20, 40))$, medium $([40, 60))$, good $([60, 80))$, excellent $([80, 100)$ and ideal state $(100)$. As can be seen from Table 8, all indicators are below the level "good", mainly in the medium range. It means that Nantong University is in the development stage. To put it differently, Nantong University's overall strength needs to be further improved.

### 4.3. Diagnosis of Obstacle Factors for HQD Level

In order to analyze the obstacle factors influencing the HQD level of schools, the obstacle diagnosis model according to Formula (13) is established and accurate measurement of all 11 indicators is required. Based on the calculation result of the HOQ level in 2020, the obstacle degrees of 11 indicators affecting the HQD level of schools are obtained as shown in Table 9. Table 9 shows that there are large differences in the degree of limitation to the HQD level among the different dimensions.

**Table 9.** The main obstacles and the order of obstacle degree affecting the HQD level of schools in 2020.

| | $C_1$ | $C_2$ | $C_3$ | $C_4$ | $C_5$ | $C_6$ | $C_7$ | $C_8$ | $C_9$ | $C_{10}$ | $C_{11}$ | Sum |
|---|---|---|---|---|---|---|---|---|---|---|---|---|
| $B_1^1$ | 0 | 0 | **0.1329** | 0 | 0.0800 | 0 | 0 | 0 | 0 | **0.0827** | 0 | 0.2956 |
| $B_2^1$ | 0.0358 | 0 | 0 | 0 | 0 | 0 | 0 | 0 | 0 | 0 | 0 | 0.0358 |
| $B_3^1$ | 0 | 0 | 0 | 0 | 0 | 0 | 0 | 0 | 0 | 0 | 0 | 0 |
| $B_4^1$ | 0.0218 | 0.0321 | 0 | 0.0543 | 0.0162 | 0 | 0 | 0.0256 | 0.0282 | 0.0287 | 0 | 0.2071 |
| $B_5^1$ | 0.0765 | 0 | **0.0777** | **0.1523** | **0.0911** | **0.0934** | **0.1564** | **0.0898** | **0.0792** | 0.0804 | **0.0720** | **0.9687** |
| $B_6^1$ | 0.0765 | 0 | **0.0777** | 0 | **0.0911** | **0.0934** | 0 | 0 | 0.0792 | 0 | **0.0720** | 0.4898 |
| $B_7^1$ | 0 | 0 | **0.1119** | 0 | **0.1312** | **0.1345** | 0 | 0 | 0 | **0.1158** | **0.1037** | **0.5971** |
| $B_1^2$ | 0 | 0 | 0 | 0 | 0 | 0 | 0 | 0 | 0 | 0 | 0 | 0 |
| $B_2^2$ | **0.1290** | 0.0021 | **0.1065** | **0.1909** | 0.0373 | 0 | 0.0267 | 0.0167 | **0.1172** | **0.1602** | 0.0123 | **0.7988** |
| $B_3^2$ | 0 | 0 | 0 | 0 | 0 | 0 | 0 | 0 | 0 | 0.0334 | 0 | 0.0334 |
| $B_4^2$ | 0 | **0.1732** | 0.0748 | 0 | 0 | 0 | **0.1506** | 0 | **0.0763** | 0 | 0.0693 | **0.5443** |
| $B_5^2$ | 0 | **0.1243** | 0 | 0 | 0.0292 | 0 | 0.0639 | **0.0896** | 0 | 0 | 0 | 0.3070 |
| $B_6^2$ | 0 | 0 | 0 | 0 | 0 | 0 | 0 | 0 | 0 | 0 | 0.0656 | 0.0656 |
| $B_7^2$ | **0.0902** | **0.1000** | 0 | 0.0534 | 0.0343 | **0.0955** | **0.1958** | 0.0707 | 0.0285 | 0.0408 | 0.0284 | **0.7378** |
| $B_8^2$ | 0.0708 | 0.0554 | 0.0479 | **0.0940** | 0.0562 | 0.0864 | 0 | 0.0277 | **0.0977** | 0.0248 | **0.0888** | **0.6496** |
| $B_1^3$ | **0.0772** | **0.1815** | **0.0784** | **0.1538** | **0.0919** | **0.0943** | **0.1578** | **0.0907** | 0.0799 | **0.0811** | **0.0727** | **1.1594** |
| $B_2^3$ | 0 | 0 | 0 | 0 | 0 | 0 | 0 | 0 | 0 | 0 | 0 | 0 |
| $B_3^3$ | 0 | 0 | 0 | 0 | 0 | 0.0551 | 0 | 0.0347 | 0 | 0 | 0 | 0.0899 |
| $B_4^3$ | 0.0002 | 0.0033 | 0.0213 | 0.0144 | 0 | 0 | 0 | 0.0415 | 0.0071 | 0 | 0 | 0.0879 |
| $B_5^3$ | 0.0241 | 0.0199 | 0 | 0.0131 | 0.0218 | 0 | 0.0492 | 0 | 0.0231 | 0 | 0.0326 | 0.1837 |
| $B_6^3$ | 0.0168 | 0.0285 | 0.0226 | 0.0527 | 0 | 0.0431 | 0.0153 | 0.0167 | 0.0157 | 0.0371 | 0.0210 | 0.2696 |
| $B_7^3$ | 0 | 0.1187 | 0 | 0 | 0.0225 | 0 | 0 | 0.0629 | 0.0131 | 0.0066 | 0.0475 | 0.2712 |
| $B_8^3$ | 0.0080 | 0 | 0.0131 | 0 | 0.0473 | 0 | 0 | 0.0255 | 0.0093 | 0.0516 | 0 | 0.1549 |
| $B_1^4$ | **0.1095** | 0 | 0.0635 | **0.1452** | 0 | 0.0763 | 0 | 0 | 0.0453 | **0.1150** | 0 | **0.5548** |
| $B_2^4$ | **0.1248** | **0.1114** | 0 | 0.0416 | 0.0599 | 0.0680 | 0 | 0.0476 | **0.1038** | 0.0317 | 0 | **0.5888** |
| $B_3^4$ | 0.0019 | 0 | 0.0323 | 0 | **0.0868** | 0.0126 | 0 | 0.0499 | 0.0511 | 0.0499 | 0 | 0.2845 |
| $B_4^4$ | 0.0139 | 0.0326 | 0.0141 | 0.0276 | 0.0165 | 0.0169 | 0.0283 | 0.0163 | 0.0143 | 0.0146 | 0.0130 | 0.2081 |
| $B_5^4$ | 0 | 0 | 0.0741 | 0 | **0.0868** | 0 | 0 | **0.0856** | 0.0755 | 0 | 0.0686 | 0.3907 |
| $B_6^4$ | 0.0398 | 0.0170 | 0.0141 | 0 | 0 | 0 | 0.0090 | **0.1427** | 0.0179 | 0.0026 | **0.1134** | 0.3565 |
| $B_7^4$ | 0.0077 | 0 | 0.0017 | 0.0003 | 0 | 0.0169 | 0.0103 | 0.0061 | 0.0018 | 0.0020 | 0.0098 | 0.0566 |
| $B_8^4$ | 0.0756 | 0 | 0.0352 | 0.0063 | 0 | **0.1137** | **0.1366** | 0.0597 | 0.0359 | 0.0409 | **0.1091** | **0.6129** |

Moreover, we can obtain the following analysis results.

(1)　With the indicators $B_3^1$, $B_1^2$ and $B_2^3$, all the obstacle degrees of each school are 0 through factor analysis, indicating that these indicators do not hinder the HQD level. Based

on the sum of obstacles for each indicator, the top 10, in descending order, are $B_1^3$, $B_5^1$, $B_2^2$, $B_7^2$, $B_8^2$, $B_8^4$, $B_7^1$, $B_2^4$, $B_1^4$, $B_4^2$.

(2) Looking closely at the top five impact factors and impacts in each school, an analysis of the number of occurrences found that $B_5^1$ (number of times is 8), $B_7^1$ (number of times is 5), $B_2^2$ (number of times is 5) and $B_1^3$ (number of times is 10) are the most common obstacle factors. This indicates that these factors have the most prominent influence among the 11 basic indicators.

*4.4. Suggestions*

Use the comprehensive evaluation model and the obstacle diagnosis model to conduct quantitative analysis on the HQD level of schools of Nantong University; the situation of the HQD level in various schools in Nantong University are objectively presented. On this basis, the corresponding policy suggestions for improving the overall level of Nantong University are put forward.

**1. University perspective.**

At present, the comprehensive strength of Nantong University is not very strong. As a local university, Nantong University must set its strategic goals around the tenet that "learning must be expected to be used and the use must be suitable for its local community". On the one hand, combining the current situation and its characteristics, Nantong University should formulate a development plan in line with itself. At the same time, the establishment and improvement of the work of the incentive mechanism mobilizes the enthusiasm of each school. On the other hand, the repeated COVID-19 outbreak has had a great impact on the development of Nantong University. For now, the impact is inevitable. Despite this, challenges and opportunities coexist. All universities are facing unprecedented challenges during the epidemic. Nantong University needs to actively cope with the negative impact, in depth analysis of the development dilemma under the epidemic, accurately find a breakthrough and take the lead in adversity to open a new bureau, so as to seize the opportunity for development.

**2. Administrative department perspective.**

First of all, the administrative departments should create a scientific, comprehensive and dynamic evaluation system of commonalities and characteristics. That is to say, the evaluation system not only considers the common requirements for the HQD of schools, but also takes into account the individual development of different types of schools.

Secondly, through regular obstacle factor analysis, administrative departments need to dynamically adjust the assessment indicators, clear the assessment direction and find out the key points of the university's development.

Additionally, each administration needs to go into all schools and understand the problems they face around the requirements of the school. Meanwhile, experts inside and outside the university are invited to help the schools analyze problems and define the development strategy.

Finally, the administrative departments should help formulate implementation plans in line with the development status of the schools and university. During the implementation of the program, a tracking mechanism should be established to keep an eye on everything and solve obstacles of schools in time.

**3. School perspective.**

Each school has its weaknesses and strengths. Thus, a school must work hard to strengthen areas of weakness and strengthen its strengths. The specific method for addressing a school's dilemmas and determining solutions includes the following points.

(1) On the one hand, comprehensively analyze the advantages and disadvantages of the school. On the other hand, the barriers that limit the development of the school at each stage are obtained through obstacle analysis.

(2) Under the current pandemic situation, the school must change its ideas, innovate its working methods and develop reform plans based on the current situation of the school.

(3) Classify all types of personnel in the school. Then, the indicators are matched with the group conditions, so as to formulate a task list that conforms to the characteristics of individuals or groups.

(4) At the same time, the potential and role of all kinds of people in the school can be fully given play through the establishment of school-level assessment or performance mechanisms. Therefore, the completion of all targets can be steadily pushed forward.

## 5. Conclusions

In order to promote economic development and enable people to live better life, the Chinese government put forward a new concept of HQD. Considering the relationship between higher education and national development, we combined the requirements of university development to put forward the connotation of HQD of schools. Based on this, an integrated framework containing a comprehensive evaluation model and obstacle analysis model of schools in the university was constructed. Firstly, due to the advantages of IFSs, this study selected the IFSs to describe the preferences of experts and indicators and determined the weights of experts and indicators. Then, the synergistic degrees of the HQD level were synthetically evaluated. Secondly, the obstacle degree model was used to find the main obstacle factors affecting the schools' improvement of the HQD level. Finally, a numerical example was addressed for a comprehensive evaluation and obstacle analysis was employed to facilitate the understanding of the proposed model. Meanwhile, through the analysis of the data, strategies and suggestions for the university to improve the HQD level were put forward. Results provided a scientific basis to allow the university to objectively know the development status of each school. The results indicated that the development directions of each school differed from each other. Therefore, the university needs to consider the commonness and individuality of the schools, formulate policies and implement programs to promote the HQD of each school, so as to improve the comprehensive strength of the university.

Obviously, there are still some drawbacks to the method proposed in this paper. Possible dependencies between indicators are not fully considered. Meanwhile, there are no indicators to measure the growth degree of each school over a developmental period, which can reflect the growth increment and development quality of the school. So, we will further concentrate on the method to enrich the theoretical system of higher education evaluation.

**Funding:** This work was supported by the General Project of Philosophy and Social Science Research in Universities of Jiangsu Province under Contract Nos. 2020SJA1594.

**Institutional Review Board Statement:** Not applicable.

**Informed Consent Statement:** Not applicable.

**Data Availability Statement:** Data sharing not applicable. No new data were created or analyzed in this study. Data sharing is not applicable to this article.

**Conflicts of Interest:** The author declares no conflict of interest.

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
