# Peer review of "Evaluation of Schools of University and Analysis of Obstacle Factors under the Background of High-Quality Development"

_sustainability, doi:10.3390/su15043754_

Round 1
Reviewer 1 Report
The author has studied the "Evaluation of schools of university and analysis of obstacle factors under the background of high-quality development" using model approaches. Although the work is well presented, there are a few areas that need to be addressed.
1. Data sources need more explanations and detail.
2. Figure 1 has no title on the vertical axis and Figures titles should be under the figures and be consistent. Fig 1 is on top of the figure and Fig. 2 is below the figure.
3. Table 7 should be table 6 and the rest of the tables numbered accordingly.
4. References need more addition with emphasis on updated relevant references and journals.
5. The entire manuscript needs to be revised by a native English speaker as there are several errors and grammatical mistakes in the manuscript. Eg, Page 1 line ----To address-----------,
page 2 line 15,--There' no denying ----------------, etc
Author Response
Dear Reviewer:
Thank you very much for your comments concerning my manuscript entitled “Evaluation of schools of university and analysis of obstacle factors under the background of high-quality development” (Manuscript ID: sustainability-2203821). Those comments are valuable and very helpful for revising and improving my paper, as well as the important guiding significance to my future researches.
I have studied the comments carefully and made corrections correspondingly on the original manuscript. A revised manuscript with the correction sections blue marked was attached as the “attachment” for easy check/editing purpose.
I tried my best to improve the manuscript and made some changes in the manuscript. These changes will not influence the content and framework of the paper. I appreciate for your warm work earnestly, and hope that the revised manuscript will meet with approval.
Once again, thank you very much for your comments and suggestions.
Sincerely yours,
Huiping Chen

Reviewer 2 Report
The paper proposes an integrated framework for HQD evaluation and obstacle factor
analysis for schools. The vagueness and imprecision are modeled through intuitionistic fuzzy sets (IFSs).
The Introduction is extensive and gives a literature overview on the subject. The cited sources are all relevant but most of the references are to papers of Chinese researchers. There are more recent works by non-Chinese researchers on evaluation of universities where the uncertainty is modeled through IFSs. For example, in the paper
Niksa-Rynkiewicz, Tacjana. Application of Intuitionistic Fuzzy Sets to the assessment of technical university students. Zeszyty Naukowe Wydziału Elektrotechniki i Automatyki Politechniki Gdańskiej, No 65, 2019, 101-103.
IFSs are used to evaluate students and a classification algorithm is proposed.
While in the paper
Bureva, V., Sotirova, E., Atanassov, K., Andonov, V. Applications of the InterCriteria analysis on the ratings of the universities in Bulgaria for 2019-2020. Annual of Informatics Section, Union of Scientists in Bulgaria, Vol. 10, 2019/2020, 14-29.
a novel approach based on IFSs is proposed for evaluation of universities and determination of possible dependencies between the evaluation criteria.
I recommend to the authors to include such recent non-Chinese sources in the Introduction which are very closely related to the topic.
In the definition of the notion of IFS (Definition 1), the expression X={x1, x2, …, xn} should be substituted by X, since in the present form the definition implies that IFS sets are finite sets.
The notion of intuitionistic fuzzy number (IFN) is used right after Definition 1 without defining it. It is important to give a definition because in the literature in recent years different concepts have been referred to as IFN.
The procedure for evaluation of experts’ weights and indicator is described in Section 3. 1 and a normalized matrix with scores is obtained. The approach is sound and similar approaches have been used for evaluation of experts in experts systems.
In Section 4, a case study is performed using data from the Ministry of Education of China. To the linguistic terms evaluations of the experts are juxtaposed IFNs. Then the importance of the first and second layers are evaluated. The captions of Table 4 and Table 5 are the same – the second needs to be corrected.
HQD of schools is evaluated for 2019-2021 using 7 levels. Then the main obstacles and the order of obstacles degree affecting the HQD level of schools in 2020 are detected.
The conclusions are supported by the results presented in the paper. Overall, the proposed approach is sound but the question of possible dependencies between the indicators should be addressed.
Author Response
Dear Reviewer:
Thank you very much for your comments concerning my manuscript entitled “Evaluation of schools of university and analysis of obstacle factors under the background of high-quality development” (Manuscript ID: sustainability-2203821). Those comments are valuable and very helpful for revising and improving my paper, as well as the important guiding significance to my future researches.
I have studied the comments carefully and made corrections correspondingly on the original manuscript. The point-by-point responses to the your comments were listed on the next page. A revised manuscript with the correction sections blue marked was attached as the “attachment ” for easy check/editing purpose.
I tried my best to improve the manuscript and made some changes in the manuscript. These changes will not influence the content and framework of the paper. I appreciate for your warm work earnestly, and hope that the revised manuscript will meet with approval.
Once again, thank you very much for your comments and suggestions.
Sincerely yours,
Huiping Chen
Responses to the comments:
Question 1:The Introduction is extensive and gives a literature overview on the subject. The cited sources are all relevant but most of the references are to papers of Chinese researchers. There are more recent works by non-Chinese researchers on evaluation of universities where the uncertainty is modeled through IFSs. I recommend to the authors to include such recent non-Chinese sources in the Introduction which are very closely related to the topic.
Response: Thanks for your comment and advice cordially. I have added some updated relevant references and journals, see the attachment for details.
Question 2: In the definition of the notion of IFS (Definition 1), the expression X={x1, x2, …, xn} should be substituted by X, since in the present form the definition implies that IFS sets are finite sets.
Response: This comment is highly appreciated and I accept the suggestion. I have changed the content of the definition, as follows:
Question 3: The notion of intuitionistic fuzzy number (IFN) is used right after Definition 1 without defining it. It is important to give a definition because in the literature in recent years different concepts have been referred to as IFN.
Response: In some papers, IFN is sometimes referred to as an intuitionistic fuzzy value. See Reference 30 for details. I referred to the statement in the paper “Chen, H. P., Xu, G. Q. Group decision making with incomplete intuitionistic fuzzy preference relations based on additive consistency. Computer & Industrial Engineering, 2019, 135: 560-567. “
Question 4: In Section 4, a case study is performed using data from the Ministry of Education of China. To the linguistic terms evaluations of the experts are juxtaposed IFNs. Then the importance of the first and second layers are evaluated. The captions of Table 4 and Table 5 are the same – the second needs to be corrected.
Response: I am very sorry for my negligence, and the error has been corrected.
Question 5:The conclusions are supported by the results presented in the paper. Overall, the proposed approach is sound but the question of possible dependencies between the indicators should be addressed.
Response: This comment is highly appreciated. In the future, I will solve this problem. So, I have added the following in the paper.
Obviously, there are still some drawbacks to the method proposed in this paper. Possible dependencies between indicators are not fully considered. Meanwhile, there are no indicators to measure the growth degree of each school over a developmental period, which can reflect the growth increment and development quality of the school. So, we will further concentrate on the method to enrich the theoretical system of higher education evaluation.
